# The Iatrogenic Development of an Anterior Cerebral Artery Pseudoaneurysm during Lamina Terminalis Fenestration–Genesis, Diagnosis and Therapy: Lessons Learned

**DOI:** 10.3390/brainsci10060357

**Published:** 2020-06-09

**Authors:** Bartoš Robert, Lodin Jan, Hejčl Aleš, Sameš Martin, Cihlář Filip

**Affiliations:** 1Department of Neurosurgery, J. E. Purkyne University, Masaryk Hospital, 401 13 Ústí nad Labem, Czech Republic; robert.bartos@kzcr.eu (B.R.); jan_lodin@hotmail.com (L.J.); martin.sames@kzcr.eu (S.M.); 2Institute of Anatomy, 1st Medical Faculty, Charles University, 128 00 Prague, Czech Republic; 3International Clinical Research Center, St. Anne’s University Hospital, 656 91 Brno, Czech Republic; 4Institute of Experimental Medicine, Academy of Sciences of the Czech Republic, 142 20 Prague, Czech Republic; 5Department of Radiology, J. E. Purkyne University, Masaryk Hospital, 401 13 Ústí nad Labem, Czech Republic; filip.cihlar@kzcr.eu

**Keywords:** intracranial pseudoaneurysm, brain edema, external ventricular drainage, lamina terminalis

## Abstract

Intracranial pseudoaneurysms (PSA) are scarcely presented in the literature. We describe the case of an intracranial PSA on the right anterior cerebral artery, which developed during the complicated surgical treatment of a ruptured right middle cerebral aneurysm. The pseudoaneurysm grew over time and was co-incidentally diagnosed 3 months after the original surgery. The PSA was successfully treated by coiling. In cases of vascular injuries during complicated brain surgery, the timely and careful radiological diagnosis of such a lesion is necessary to allow its fast and proper treatment and thus prevent the patient from potential risks.

## 1. Introduction

Intracranial pseudoaneurysms (PSA) are rare vascular entities which occur in cases of head trauma, vascular collagenopathies such as Marfan’s syndrome, infection, vasculitis or aneurysm rupture. Furthermore, they can be iatrogenic, developing after endoscopic surgery in the sellar area, endovascular procedures of intracranial vessels, external ventricular drain insertion or even a rare case of PSA occurrence following a transcallosal approach. Generally speaking, literature concerning PSA occurrence following aneurysm surgery is, for the most part, discreetly silent. It is certainly a rare complication of these procedures. We found only three cases of pseudoaneurysm development in association with aneurysm surgery in the current literature; all three of them were treated surgically [1,2,3]. On the other hand, various types of endovascular therapy for PSA are available. The deconstructive treatment of a PSA by occluding the injured artery requires sufficient collateral flow [4]. Reconstructive treatment means closing the site of the injury with coils while preserving the parent artery, sometimes utilizing remodeling tools such as a balloon, a stent or a flow diverter [5,6]. The use of a stent or a flow diverter requires long-time use of anti-platelet treatment. In addition, reconstructive treatment is associated with a higher risk of recanalization of the injured site.

In this paper, we describe the iatrogenic development of an anterior cerebral artery (ACA) PSA which occurred during a technically difficult procedure of middle cerebral artery aneurysm clipping, as well as the successful concurrent endovascular treatment of the PSA.

## 2. Case Report

A 49-year-old man was admitted to his local neurological department in September of 2016 with symptoms of aphasia, aggressiveness and confusion. A computed tomography (CT) and computed tomography angiography (CTA) were performed and showed a subarachnoid hemorrhage with hemocephalus (Fisher 4) as well as an aneurysm of the right middle cerebral artery (MCA) bifurcation (M12). Due to deterioration of consciousness, the patient was sedated, intubated and transported to our neurosurgical department for further treatment. After consultation, the patient was indicated for acute surgical clipping of the aneurysm by the surgeon on duty (the first author). A standard pterional craniotomy and durotomy were performed; however, major brain edema with brain herniation through the craniotomy was present and did not allow cisternal dissection. Antiedema measures, including lumbar drainage of the cerebrospinal fluid (CSF), were unsuccessful; therefore, the surgeon decided to utilize major retraction of the frontal lobe in order to quickly gain access to the lamina terminalis, aiming to fenestrate it. Unfortunately, this maneuver resulted in arterial bleeding from this area, which was successfully controlled with Surgicel, and fenestration of the lamina terminalis was microsurgically performed. This resulted in sufficient CSF decompression and brain relaxation, which allowed the dissection of the carotid and Sylvian cisterns and resulted in the successful clipping of the ruptured aneurysm. When leaving the surgical field, the surgeon did not observe any bleeding in the anterior cerebral artery area; the original source bleeding was controlled with a Surgicel tamponade. A CT and CTA were performed on the first postoperative day (Figure 1A) and showed no complications.

Postoperatively, the patient developed severe vasospasms, which were successfully treated pharmacologically, including with a one-time pharmacological angioplasty with milrinone. On the fifth postoperative day, a final CTA was performed and demonstrated successful clipping of the ruptured aneurysm without any other vascular anomalies apart from a previously described small mirror aneurysm on the left (Figure 1B). On the 18th postoperative day, the patient was transferred to his local hospital for rehabilitation in good clinical condition—Glasgow coma scale 15, lucid, oriented to place, person and time and a residual minor left-sided hemiparesis accented on the upper limb. Following the transfer, the patient’s clinical condition slowly deteriorated; he became disoriented and showed signs of bradypsychia. A follow-up CT performed 3 months after the surgery showed widening of the ventricles as well as a pseudoaneurysm (PSA) in the distal A1 anterior cerebral artery (ACA) segment (Figure 1C).

We decided to treat the pathologies immediately, which necessitated the insertion of a caval filter due to deep vein thrombosis. The following day, a ventriculo-peritoneal shunt (Certas, Medtronic) was implanted and, six days later, diagnostic angiography was performed and verified the presence of a 17 mm PSA in the distal A1 segment of the ACA (Figure 2A,B). The PSA was coiled the following day (Cosmos Complex spirals into the aneurysm dome, Target Nano in the aneurysm ostium and the A1 segment of the ACA). The patency of the ipsilateral A2-3 via the anterior communicating artery was verified in the final angiogram, which also demonstrated complete occlusion of the PSA (Figure 2C,D). No further neurological deficits developed following the endovascular procedure and the patient was successfully treated for suspected shunt meningitis (negative bacterial cultures) with meropenem. The patient was transferred to his local hospital and later released.

Twenty-two months later, he was readmitted with a severe headache; a CTA was performed and did not show PSA recurrence or a subarachnoid hemorrhage but identified thrombosis of the straight, right sigmoid, right transverse and partial left transverse sinuses. Anticoagulation therapy was initiated and resulted in the recanalization of all sinuses, with the exception of the right transverse sinus. Afterwards, the patient and his family elected surgical clipping of the contralateral mirror MCA aneurysm, which was performed by the original surgeon 11 months later. The final procedure was uncomplicated (Figure 3A,C), and the patient was released with a minor organic psychosyndrome, completely self-dependent, without hemiparesis.

## 3. Discussion

Iatrogenic pseudoaneurysms can occur as rare complications of several neurosurgical procedures, including transsphenoidal surgery [7,8], external ventricular drain insertion [9,10] or stereotactic radiosurgery of cerebellopontine angle lesions [11,12]. The genesis of these lesions during open neurosurgical procedures is sparsely discussed in the literature, perhaps due to their rarity or due to the fact that they are traumatic events for the operating surgeons, not suitable for publishing.

Dunn et al. [13] describe the development of a pericallosal artery PSA a year after the transcallosal surgery of an optic glioma in a child. The PSA was later resected, and the artery wall defect was repaired using a graft from the superficial temporalis artery. However, this report does not depict PSA formation during a primarily vascular procedure—aneurysm clipping. A historical description of such a situation is presented by Cosgrove et al. 1983 [1], who describe the formation and rupture of a pseudoaneurysm in close proximity to a clipped anterior communicating artery aneurysm 21 days after the primary surgical procedure. According to the authors, the primary procedure was in no way complicated; however, they suggest possible trauma to a small branch of the ACA or direct trauma to the ACA during intraoperative hypotension. The PSA was successfully treated surgically via an additional clip. Eleven years later, Rowed et al. [2] presented a case of iatrogenic trauma to the internal carotid artery (ACI) during the clipping of a ruptured aneurysm of the basilar and superior cerebellar artery angle. The traumatic ACI lesion spontaneously stopped bleeding and was therefore not directly treated during the primary procedure. This scenario most resembles our own case. Eight days after the primary procedure, the rupture of a related ACI pseudoaneurysm occurred and a revision procedure requiring the sacrifice of the ACI was performed. This unfortunately resulted in a permanent neurological deficit. In hindsight, the authors suggest more careful observation of the trauma site as well as preventively suturing the damaged artery wall. The surgical treatment, however, led to a permanent neurological deficit which emphasizes the role of endovascular therapy as the first-line treatment of such a PSA unless a bypass is needed.

A more recent case report is presented by Shoja et al. [3], who describe the formation of a PSA 18 days after the uncomplicated surgical clipping of an anterior communicating artery (AComA) aneurysm, which was preceded by an unsuccessful coiling attempt. A temporary clip was used during the surgical procedure. The PSA formed again on the AComA but in an opposite direction to the original aneurysm (superior direction). An attempted endovascular procedure was unsuccessful, and the PSA was finally treated surgically with a clip via the original approach. The authors suggest several possible reasons for PSA formation, including trauma during the first endovascular procedure, residual aneurysm after clipping or ischemia of the artery wall due to the obliteration of the vasa vasorum by the surgical clip. The authors consider direct trauma to the AComA or its branches to be the least probable explanation of PSA formation. Our case is perhaps most similar to the case described by Rowed et al., involving PSA formation as a direct surgical complication, which thankfully resulted in a good outcome for the patient due to the effective endovascular treatment of the PSA [14,15]. It is necessary to accent the fact that, in our case, reaching the lamina terminalis in conditions of extreme brain edema was a critical step in achieving sufficient brain relaxation and thus successful clipping of the ruptured aneurysm. Unlike the above mentioned case reports, the development of a PSA was thankfully diagnosed as a coincidental finding on a follow-up CT, not by rupture of the lesion. This resulted in timely endovascular treatment which allowed the successful treatment of the lesion with minimum neurological morbidity.

Our department represents a medium-volume center in treating cerebral aneurysms (400 aneurysms within 8 years). This is the first time such a case has occurred, and it has led to a paradigm shift in our approach when performing the surgical clipping of a ruptured brain aneurysm in conditions of severe brain edema. We believe this will be beneficial to future patients who are treated in similar conditions. In cases of suspected injury to an arterial vessel, large or small, extreme care must be taken to visualize, control and treat the suspected lesion by means of clip or suture. Tamponade in combination with local hemostatic agents is not a sufficient treatment strategy. In cases where such treatment is not possible, an early diagnosis must be obtained after the treatment of vasospasms, as these can delay PSA formation. We prefer CTA from digital substraction angiography (DSA) in order to avoid the 1% risk of a neurological deficit associated with angiographic assessment [16]. Nonetheless, a following endovascular procedure is then the treatment method of choice.

In cases where the surgeon predicts severe brain edema on the basis of preoperative CT scans, the preoperative insertion of an external ventricular drain should be considered in cases with sufficient ventricle width. Insertion of a lumbar drain may not be sufficient in such cases. As a last resort, in cases of brain edema which is unresponsive to conventional antiedema therapy (hyperventilation, osmotherapy, lumbar drain), an emergent external ventricular drain may be inserted from the superior part of the pterional craniotomy [17], via the Park’s, Paine’s or Hyun´s point, rather than risking brain trauma by attempting to reach the lamina terminalis directly.

## 4. Conclusions

The occurrence of an intracranial PSA after the surgical repair of an aneurysm is a rare complication. In cases of extensive perioperative brain edema which is unresponsive to regular management (osmotherapy, lumbar drain, hyperventilation), emergent external ventricular drain (EVD) implantation, rather than a direct approach to the lamina terminalis, should be attained in order to avoid vascular lesion complications. In case of any suspicion, targeted diagnostic steps such as repeated CTA are to be performed in order to confirm the development of a PSA, which would facilitate early treatment.

## Figures and Tables

**Figure 1 brainsci-10-00357-f001:**
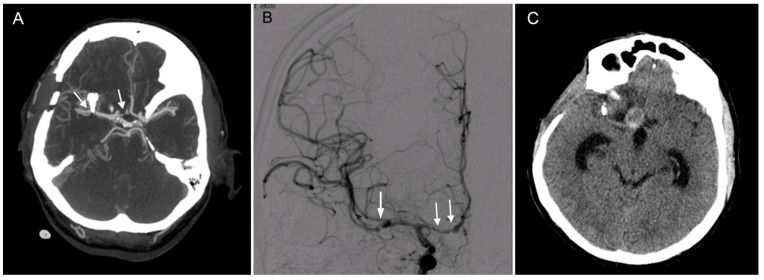
Clipping of a ruptured middle cerebral artery (MCA) aneurysm complicated in the postoperative course by vasospasm and the development of a pseudoaneurysm (PSA). (**A**). A computed tomography angiography (CTA) performed on the first postoperative day shows clips after treating an MCA aneurysm (left arrow) and a narrowing of the right A1 segment (right arrow), which eventually led to later complications. (**B**). Postoperatively, the patient developed a severe vasospasm (white arrows) which was successfully treated including an intra-arterial pharmacological angioplasty with milrinone. (**C**). Brain computed tomography (CT) performed 3 months later presenting hydrocephalus and a suspicious PSA of the distal right A1 segment.

**Figure 2 brainsci-10-00357-f002:**
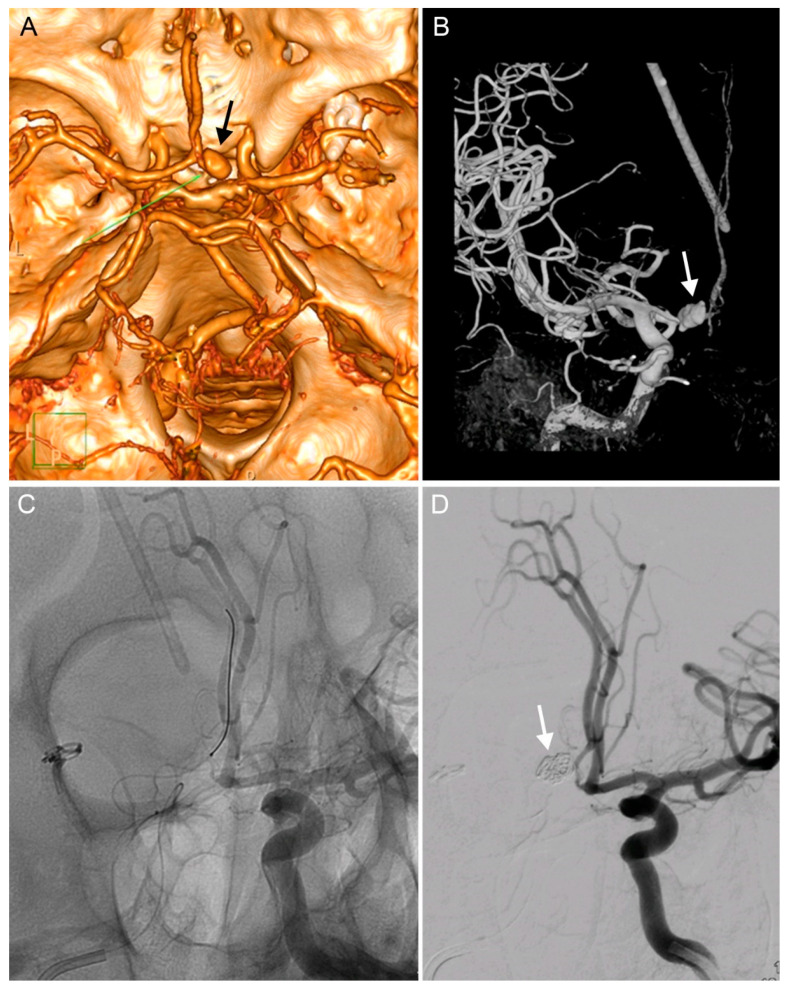
Diagnosis and endovascular treatment of an iatrogenic PSA of the A1 segment of the right anterior cerebral artery (ACA). (**A**). CT angiography showing a PSA of the right A1 segment (arrow). (**B**). The same PSA on a diagnostic digital substraction angiography (DSA, arrow) after implanting a ventriculo-peritoneal (VP) shunt. (**C**). An endovascular intervention for the PSA: a guidewire in the right ACA. (**D**). Angiography showing completed coiling of the PSA and showing patency of both A2 segments.

**Figure 3 brainsci-10-00357-f003:**
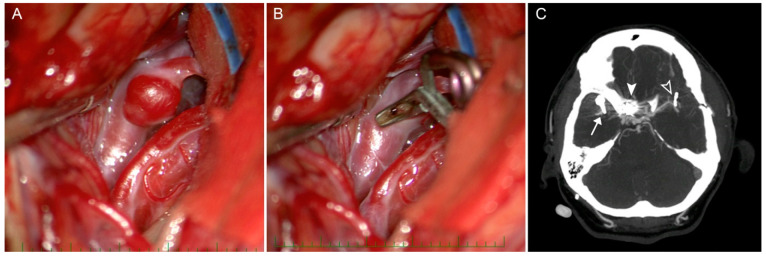
Clipping of an unruptured left M2 aneurysm. (**A**). Intraoperative view of the left unruptured M2 aneurysm. (**B**). Intraoperative view after successful clipping the M2 aneurysm. (**C**). CTA after final treatment: a clip of a ruptured right M12 bifurcations (arrow), coiling of the right distal A1 (full white arrowhead) and a clip of unruptured left M2 (empty arrowhead).

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
