# Peer review of "The Iatrogenic Development of an Anterior Cerebral Artery Pseudoaneurysm during Lamina Terminalis Fenestration–Genesis, Diagnosis and Therapy: Lessons Learned"

_brainsci, 2020, doi:10.3390/brainsci10060357_

Round 1

Reviewer 1 Report

This manuscript details a well-controlled case study to investigate iatrogenic pseudoaneurysm and how repeated computed tomography angiography could serve as diagnostic tool and facilitate therapy. The findings of the case study are interesting and important, however there are some minor issues that need to be addressed in the manuscript

  • In the introduction, it would be greatly helpful if the authors add other endovascular options that are available for the treatment of iatrogenic ACA injuries.
  • In the case report, add a sentence about the risk of neurologic deficit associated with repeated angiographic assessment.
  • Include the pre-surgical assessment using angiography as a Figure 1. This would give a better picture of the lesions before surgery.

Author Response

Dear reviewer,

thank you very much for your kind response. The comments are very reasonable and we edited the manuscript as suggested in all your comments.

  • In the introduction, it would be greatly helpful if the authors add other endovascular options that are available for the treatment of iatrogenic ACA injuries.

The following text was added::

On the other hand, various types of endovascular therapy for PSA are available. Deconstructive treatment of a PSA by occluding the injured artery requires sufficient collateral flow (4). Reconstructive treatment means closing the site of injury with coils while preserving the parent artery sometimes utilizing remodelling techniques, such as a balloon, a stent or a flow diverter (5,6). The use of a stent or a flow diverter requires long-time use of anti-platelet treatment. Also, reconstructive treatment is associated with a higher risk of recanalization of the injured site.

  • In the case report, add a sentence about the risk of neurologic deficit associated with repeated angiographic assessment.

We added the following sentence:

We prefer CTA from DSA to avoid the 1% risk of a neurological deficit associated with angiographic assessment.

Willinsky RA, Taylor SM, TerBrugge K, Farb RI, Tomlinson G, Montanera W. Neurologic complications of cerebral angiography: prospective analysis of 2,899 procedures and review of the literature. Radiology. 2003 May;227(2):522-8. doi: 10.1148/radiol.2272012071. Epub 2003 Mar 13.PMID: 12637677Review.

  • Include the pre-surgical assessment using angiography as a Figure 1. This would give a better picture of the lesions before surgery.

The pre-treatment CTA was from a local hospital and unfortunatelly the quality of the images was not adequate for publication.  

Reviewer 2 Report

This study present a case study on intracranial pseudoaneurysms. The study is well designed and clearly described.

I only have some minor concerns on the introduction and discussion sections. It would be great if the authors can provide additional information on the research background, i.e. what has been done in this field. In addition, the new findings could be further elaborated to demonstrate how this study differs from previous ones, or, what observed in this study may support previous findings.

Author Response

Dear reviewer,

thank you very much for your kind response. The comments are very reasonable and we edited the manuscript as suggested in all your comments.

This study present a case study on intracranial pseudoaneurysms. The study is well designed and clearly described.

I only have some minor concerns on the introduction and discussion sections. It would be great if the authors can provide additional information on the research background, i.e. what has been done in this field. In addition, the new findings could be further elaborated to demonstrate how this study differs from previous ones, or, what observed in this study may support previous findings.

We found only 3 cases of PSA development after brain aneurysm surgery in the literature and all were treated surgically. The case most resembling ours in the scenario was treated by trapping the parent artery which resulted in a permanent neurological deficit. This leads to preference of endovascular therapy as the first-line treatment of PSA as the first line treatment unless a bypass is necessary.  – this is further described in the Discussion section